# The IUCN Red List and newspaper coverage of threatened freshwater eel species in Japan: A variable but limited influence

Kenzo Kaifu[1]*, Hiromi Shiraishi[1], Atsushi Ishii[2], Aoi Sugimoto[3]

**1** Chuo University, 1-4-1 Otsuka, Bunkyo-ku, Tokyo, Japan, **2** Center for Northeast Asian Studies, Tohoku University, 41 Kawauchi, Aoba-ku, Sendai, Miyagi, Japan, **3** Graduate School of System Design and Management, Keio University, 4-1-1 Hiyoshi, Kouhoku-ku, Yokohama-shi, Kanagawa, Japan

* kkaifu001t@g.chuo-u.ac.jp

## Abstract

The IUCN Red List is a vital tool for identifying threatened species and raising public awareness. This study examined how the listing of freshwater eels (*Anguilla* spp.) influenced newspaper coverage in Japan, a major consumer of eels. We analysed 8387 eel-related articles from four major newspapers between 1992 and 2021 using text-mining and statistical methods, along with manual review of a part of the articles. Peaks in article numbers coincided with major events, such as the IUCN listing of the Japanese eel as Endangered in 2014. However, the influence of the Red List was short-lived and limited in strength, with minimal media attention given to non-native eel species such as European eel. Category-specific analysis revealed that issues related to food and trade dominated media coverage, while conservation-focused reporting was less prominent. Our findings suggest that while the Red List can momentarily increase media attention, its impact on long-term public awareness is limited. Strengthening expert engagement, international cooperation, and consumer education could be essential to enhance the conservation impact of the Red List.

## Introduction

The International Union for Conservation of Nature (IUCN) Red List of Threatened Species, established in 1964, serves as the global standard for assessing the conservation status of individual species. It plays a crucial role in biodiversity conservation by identifying species at risk of extinction and categorising them into different threat levels [1]. One of the primary functions of the IUCN Red List is to increase public awareness of species classified as threatened [2–3] (Rodrigues et al. 2006; Betts et al. 2019). By providing a globally recognised framework for categorising species based on extinction risk, it highlights conservation priorities and draws attention to species in urgent need of protection [4]. Since public concern is closely linked to the effectiveness of conservation policies [5], the role of the IUCN Red List in

**Data availability statement:** Data cannot be shared publicly because they are owned by third parties and the authors do not have permission to distribute them. These databases normally require a paid subscription for use. We accessed them through the institutional contract of Chuo University, to which some of the authors are affiliated. The data used in this study are available from the following databases: Asahi Shimbun Cross-Search: https://xsearch.asahi.com/ Maisaku Database: https://mainichi.jp/contents/edu/maisaku/ Nikkei Telecom 21: https://telecom.nikkei.co.jp/?from=logo Yomidasu Rekishikan: https://database.yomiuri.co.jp/ Fellow researchers may access these data by following the instructions provided on each database's website and obtaining a paid subscription through their institution.

**Funding:** This study was funded by Asahi Glass Foundation, Society for the Promotion of Science (JSPS) KAKENHI Grant Number JP22H00371, and Chuo University.

**Competing interests:** This study analyses the influence of the IUCN Red List on newspaper coverage. At the same time, two of the authors, Kaifu and Shiraishi, are members of the IUCN Species Survival Commission's Anguillid Eel Specialist Group (AESG) and participated in the Red List assessments of Anguilla species published in 2014 and 2020. While there are no financial conflicts of interest associated with their membership in AESG, their affiliation with the organisation that is the subject of this research is disclosed here for transparency. this does not alter our adherence to PLOS ONE policies on sharing data and materials.

emphasising public awareness is particularly significant. However, despite its importance, the extent to which the IUCN Red List influences societal attitudes and policy decisions remains a topic of ongoing research [6] (e.g., Possingham et al. 2002).

Despite its potential influence, research on the influence of the IUCN Red List on public awareness remains limited. For mammal species, for instance, Van Huynh [7] found a significant correlation between the IUCN Red List status of a species and the frequency of Google searches for that species, suggesting an increase in public interest. Betts et al. [3] also identified peaks in online search activity that could be raised by the IUCN Red List. While internet search trends provide valuable insights into public perception [7–10], they may be subject to potential biases. For example, frequent internet users may belong to specific demographic groups, or people who search for information on endangered species online may already have an interest in conservation. Interview surveys can address this issue by collecting background information on respondents, which offers deeper insights into public awareness. However, such surveys are often constrained by research resources that limit sample sizes and the geographical scale of the surveys. In contrast, media analysis provides a broader perspective, because newspaper articles and other media reports often reflect public sentiment. Examining the content and frequency of such reports can offer insights into regional trends in public awareness [11–13]. Therefore, to contribute to a better understanding of the influence of the IUCN Red List on public awareness, this study investigated newspaper coverage of threatened species listed in the IUCN Red List.

This study focuses on freshwater eels of the genus *Anguilla*. Freshwater eels, comprising 16 species (three of which are divided into subspecies), are distributed worldwide except in polar regions, the regions adjacent to the South Atlantic, and the western Americas [14]. The first IUCN Red List assessment for freshwater eels was published in 2008 for the European eel (*A. anguilla*), which has remained classified as Critically Endangered [15]. By 2019, subsequent assessments evaluated the remaining 15 species.Excluding the four categorised as Data Deficient, 12 species including the European eel have been assigned IUCN Red List statuses. Of these, 10 are classified in one of the following categories: Critically Endangered, Endangered, Vulnerable, or Near Threatened. Given that most *Anguilla* species have been classified in these categories in recent years, they provide an ideal case study for examining how red-listing has influenced newspaper coverage.

In terms of the geographic focus, our research specifically examines newspapers in Japan, where eels hold significant cultural and economic value. In Japan, eels are a highly popular food, and *unaju* or *unadon*, which consists of grilled eel (*kabayaki*) served over rice, is regarded as an important part of the country's traditional food culture (Fig 1) [16]. Freshwater eels have been consumed for over 3000 years in the Japanese archipelago [17], and more recently, according to FAO data, Japan accounted for approximately 70% of global production of *Anguilla* species (combined capture fisheries and aquaculture) [18]. Although Japan's share of the global market has declined since 2000, it still remains the centre of eel consumption [19–20]. As a result, eel-related topics are frequently reported in the Japanese media, making it feasible to conduct a quantitative analysis of newspaper articles. In support of this,

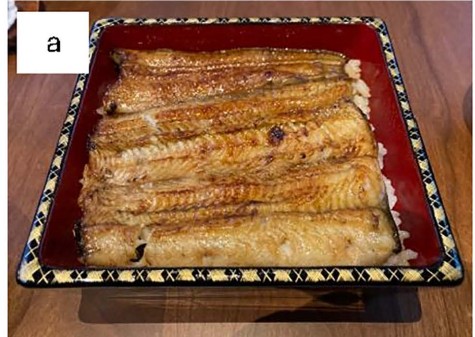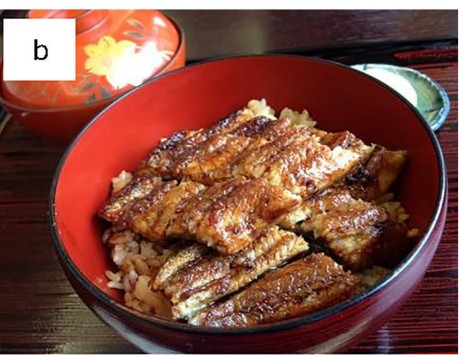

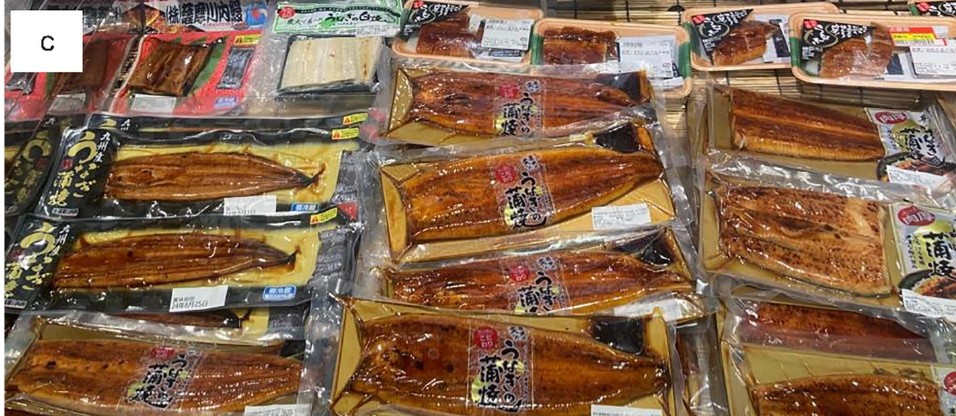

**Fig 1. Traditional Japanese-style eel dishes.** a and b, *unaju* (*kabayaki* over rice) served in traditional Japanese lacquerware; c, prepared *kabayaki* sold in a supermarket.

Sugimoto et al. [21] examined the number of inquiries made by the media and other sources to researchers at the Japan Fisheries Research and Education Agency between 2010 and 2018 and found that inquiries concerning eels were the most frequent among any fish species. Additionally, the Japanese eel (*A. japonica*), which inhabits in East Asia and consists major part of eel consumption in Japan, has been experiencing a significant population decline and has been classified as Endangered on the IUCN Red List since 2014 [22–23]. These factors make freshwater eels a compelling subject for assessing the influence of the IUCN Red List on newspaper coverage, particularly in Japan.

To contribute to a deeper understanding of the influence of the IUCN Red List on media coverage, which may further influence public awareness of threatened species, this study analyses Japanese newspaper articles and seeks to address the following questions: (1) whether the IUCN Red List listing of freshwater eels increased the number of newspaper articles, and (2) whether the IUCN Red List influenced the thematic focus (e.g., conservation awareness) of media coverage. To achieve this, we analysed newspaper articles published in Japan over the past 30 years—covering periods before and after the IUCN Red List assessments of freshwater eels. In 2017, newspaper circulation per 1,000 people in Japan was 381.4, making it the highest in the world, significantly ahead of second-place India (280.4) [24]. Given that a wide segment of the population in Japan obtains information through newspapers, the country provides an excellent setting for studying public awareness using newspaper articles. Using text-mining software, we examined trends in the number of eel-related articles and categorised their content. The number of eel-related articles was then classified based on their content, and possible relationships between numbers of articles and the years when eel species consumed in Japan were designated as Endangered or Near-Threatened were examined. Additionally, a part of the articles was randomly selected

and reviewed to evaluate their context. Through these approaches, this study provides novel insights into how Red-Listing has influenced newspaper coverage in Japan. The findings contribute to a deeper understanding of the IUCN Red List's effectiveness as a tool for raising public awareness through media reports.

## Materials and methods

### Target species

Freshwater eels (*Anguilla* spp.) are catadromous fish that spawn in offshore oceanic waters. Their leaf-like leptocephalus larvae drift with ocean currents before migrating to freshwater and estuarine habitats, where they metamorphose into 'glass eels' [25–26]. These eels spend most of their lives in continental waters for varying durations before undergoing sexual maturation. Once mature, they return to the open ocean to spawn, after which they are thought to die, exhibiting a semelparous reproductive strategy [27–29].

Deu to their significant commercial value, freshwater eels face critical conservation challenges. Although aquaculture techniques have been developed, full-cycle captive breeding remains limited to experimental settings, and commercial production still relies entirely on wild-caught glass eels [29]. Farmed eels, which constitute the majority of those consumed, are raised from juvenile eels captured in natural habitats. Consequently, glass eels are heavily harvested in many countries and are subject to extensive international trade for aquaculture [16,18]. Once farmed, eels are further distributed globally, either as live specimens or processed products. The centre of global demand for eels lies in East Asia, including Japan [19–20]. The Japanese eel, which is distributed across East Asia, has experienced a population decline and was classified as Endangered by the IUCN in 2014 [23]. The European eel was heavily exported to East Asia from the end of the twentieth century to the beginning of the twenty-first century; however, due to a significant population decline, it was listed in Appendix II of CITES in 2007 and classified as Critically Endangered by the IUCN in 2008 [15]. Following the decline of the Japanese eel and the restrictions on international trade of the European eel, demand has now shifted towards the shortfin eel (*A. bicolor*) and American eel (*A. rostrata*) [19,30], that were classified as Near Threatened and Endangered by the IUCN in 2014, respectively [31–32]. Furthermore, in the late June 2025, the European Union and Honduras have submitted a proposal to the CITES Secretariat to include all species of the genus *Anguilla* in Appendix II of the Convention. A decision on this proposal is scheduled to be made at CoP20, which will be held at the end of November 2025. The major events related to Anguilla species, such as the IUCN Red List assessments and CITES Appendix listings, are summarised in Table 1.

### Newspaper article survey

Articles were collected from the four major Japanese daily newspapers with the highest circulation: *Asahi Shimbun*, *Mainichi Shimbun*, *Nikkei*, and *Yomiuri Shimbun*. Article data were obtained from their respective digital archives: *Asahi Shimbun Cross-Search* (https://xsearch.asahi.com/), *Maisaku Database* (https://mainichi.jp/contents/edu/maisaku/), *Nikkei Telecom 21* (https://t21.nikkei.co.jp/g3/CMN0F11.do), and *Yomidasu Rekishikan* (https://database.yomiuri.co.jp/).

The search was performed using the Japanese keyword *unagi* ('eel' in Japanese) in different scripts, including *katakana*, *hiragana*, and *kanji*. Articles published between 1992 and 2021 were retrieved, with full-text searches conducted on article texts. Initially, 35,895 articles containing the term *unagi* were identified across the four newspapers. Then, to extract articles where eels were the central theme, a subset of 8387 articles was selected based on the presence of eel-related keywords in their titles. The keywords used to identify eel-related articles were: *unagi* (regardless of script type), *doyo* (summer season marker for eel consumption), *ushinohi* (midsummer eel-eating day), *shirasu* (glass eel), *kabayaki* (grilled eel), and *unadon* and *unaju* (*kabayaki* over rice). To eliminate irrelevant articles, those in which eels were not mentioned as food or a living organism in the article title were manually excluded. These included articles referencing idiomatic expressions (e.g., *unaginobori* and *unaginonedoko*) and entities whose names contain the word '*unagi*', though

**Table 1. Timeline of eel-related events in the context of resource management at both global and national scales between 1992 and 2021.**

| Year | Eel-related event | |
|---|---|---|
| | **Global** | **Domestic (Japan)** |
| 2007 | • European eel was listed on CITES Appendix II [20]<br>• The US government banned importing of eel from China after detecting the use of prohibited antimicrobial agent [33] | |
| 2008 | • European eel was classified as Critically Endangered in the IUCN Red List [15] | • The mislabeling of eel origins became a social issue [35] |
| 2009 | • CITES regulation on European eel came into force [20] | |
| 2010 | • EU banned international trade of the European eel [20]<br>• Shortfin eel was listed as Least Concern in the IUCN Red List [31] | |
| 2012 | • The US government considered proposing the inclusion of all eel species not yet listed under CITES in Appendix II [34]* | • Historical low landings of the Japanese eel glass eel [22] |
| 2013 | | • Japanese eel was listed as Endangered in the national red list of Japan [36] |
| 2014 | • European eel was re-assessed and listed as the same category (CR) in the IUCN Red List [15]<br>• European eel was listed in the Appendix II of CMS [20]<br>• American eel and Japanese eel were classified as Endangered in the IUCN Red List [23,32]<br>• Shortfin eel was listed as Near Threatened in the IUCN Red List [31] | |
| 2020 | • European eel, Japanese eel, and shortfin eel were re-assessed and listed as the same categories in the IUCN Red List [15,23,31] | |

* Subsequently, the US government decided not to submit the proposal.

they are not eel as biological organisms (e.g., personal names, geographic locations, shrines, movie titles, and fictional characters). Articles with missing content due to copyright restrictions were also removed. Finally, the selected articles (*n* = 8387) were subjected to further analysis.

For text analysis, KH Coder 3 [37] was used. Along with an analysis of the total number of articles, we counted the number of category- and species-specific articles. To classify the contextual themes of the articles for category-specific analysis, a co-occurrence network analysis was performed. This approach identifies clusters of frequently co-occurring words, allowing for the categorisation of articles into distinct thematic groups. Based on the co-occurrence network results, six themes were identified: Food, Mislabeling of Origin, Aquaculture, Resources, Fisheries, and Trade (see Results for details). To categorise the 8387 articles into these six themes, representative keywords were assigned to each category. Keywords were selected based on their high frequency within a given theme and low occurrence in others based on a review of the arbitrarily selected articles (Table 2). Since many articles covered multiple contexts, a single article could be classified into multiple categories.

Regarding species-specific analysis, there are 16 species of freshwater eels globally, with the Japanese eel being the primary species consumed in Japan. However, in recent years, other eel species have been imported and consumed in Japan, including the European eel, the American eel, and the shortfin eel [19,30,38]. To assess the relationships between newspaper coverage of these non-native eel species and their IUCN Red List status, the annual numbers of articles

**Table 2. Key words used for categorising articles.**

| Categories | Key words |
|---|---|
| Food | *taberu* (eat), *kabayaki*, *aji* (taste), *doyo* (summer season marker for eel consumption), *ushi* (mid-summer eel-eating day) |
| Mislabeling of Origin | *giso* (fraud) |
| Resource | *shigen* (resource), *hozen* (conservation), *zetsu-metsu* (extinction), *kigu* (concern), *hogo* (protect) |
| Aquaculture | *yosyoku* (aquaculture), *yoman* (eel farming), *yosyokuike* (aquaculture farm) |
| Fishery | *gyogyo* (fishery) |
| Trade | *yunyu* (import), *yusyutsu* (export), *boeki* (trade) |

The keywords are shown in Romanised Japanese, with their meanings provided in parentheses following each word. Articles containing any of the keywords were classified into the corresponding category. An article could belong to multiple categories. For 'Mislabeling of Origin' and 'Fishery', these categories have corresponding Japanese terms that match their category names one-to-one, and articles in these categories almost always include those keywords. Therefore, each of these categories has only one keyword. For 'Mislabeling of Origin', the corresponding term in Japanese is 'sanchi-giso' (fraud of origin), which consists of two words. To limit the keyword to a single word, we selected 'giso' (fraud), which has a closer connection to 'Mislabeling of Origin', as the keyword.

mentioning American eel, European eel, Japanese eel, and shortfin eel were counted. Keywords used to identify articles on specific eel species are shown in Table 3.

## Statistical analysis

Following Sampei and Aoyagi-Usui [39], who compared climate change-related events with newspaper articles, the number of newspaper articles was compared with chronological record of events related to eels. Both the total number of articles and the frequency of articles related to specific categories or eel species were examined, aligning these data with key eel-related events. Major events related to eels over the 30-year period from 1992 to 2021 were summarised in Table 1. These include international developments such as the publication of IUCN Red List assessments and the listing of species in CITES appendices, as well as several eel-related issues that were extensively reported in Japan. Among these domestic events were cases of mislabeling of eel origins, concerns over residual antibiotics, and poor catches of glass eels used for aquaculture.

Two types of analyses were conducted in this study. First, statistical peak detection was applied to identify years in which the number of articles was exceptionally high. The peak detection was performed for total, category-specific, and species-specific articles. The years when statistical peaks were detected were then compared with the timeline of eel-related events, such as publication of the IUCN Red List (Table 1). Additionally, fluctuations in article volume across different categories and species were analysed to assess their relative contributions. By combining the results of the category-specific and species-specific analyses, the possible influence of the IUCN Red List on Japanese newspaper coverage was evaluated. To identify years in which the number of articles exhibited statistically significant peaks, Rosner's Test for Outliers, a generalised form of the Extreme Studentized Deviate (ESD) Test [40], was applied. This test detects multiple peaks in a univariate dataset by identifying the most extreme data points and comparing their deviation from the sample mean against a critical threshold. Unlike standard ESD tests that assume a single outlier (due to the difference in the purpose of detection, this study uses the term 'peak' instead of 'outlier'), Rosner's Test is suitable when multiple outliers may exist within a dataset. The analysis was performed using R version 4.4.1 [41], employing the rosnerTest function from the EnvStats package. The test was applied separately to three datasets: total annual article counts, article counts by

**Table 3. Key words used to pick-up species specific articles.**

| Species | Key words |
|---|---|
| American eel | *amerika-unagi* (American eel), *rosutorata* (*Anguilla rostrata*) |
| European eel | *yoroppa-unagi* (European eel), *angira-syu* (*Anguilla anguilla*) |
| Japanese eel | *nihon-unagi* (Japanese eel), *japonika-syu* (*Anguilla japonica*) |
| Shortfin eel | *baikara* (*Anguilla bicolor*), *bikara* (*Anguilla bicolor*) |

The keywords are shown in Romanised Japanese, with their meanings provided in parentheses following each word. Articles containing any of the keywords were classified into the corresponding species. A single article could include more than one species.

category, and article counts by eel species. A maximum of three potential peaks ($k=3$) was evaluated to ensure comprehensive detection of significant deviations from expected trends.

Second, to examine whether the articles classified into six categories based on the co-occurrence network analysis, as well as the number of articles on the four eel species consumed in Japan, had a statistically significant impact on the total number of articles, regression analyses were conducted separately for category-specific and species-specific analyses. For the species-specific analysis, a linear multiple regression model was applied, with the yearly total number of articles as the response variable and the number of articles in which each of the four eel species was mentioned in each year as explanatory variables. This approach was chosen as multicollinearity was not detected among the explanatory variables. In contrast, for the category-specific analysis, multicollinearity was confirmed among the six thematic categories (Food, Mislabeling of Origin, Aquaculture, Resources, Fisheries, and Trade). To account for this, a Generalized Additive Model (GAM) was employed instead of linear regression. The response variable was the yearly total number of articles, and the explanatory variables were the numbers of articles mentioning each of the six categories in each year. A smoothing function (s()) was applied to the explanatory variables to model potential non-linear relationships. The GAM analysis was performed using the mgcv package in R version 4.4.1 [41], with optimal smoothing determined by restricted maximum likelihood (REML).

### Contextual analysis of peak years

In both the category-specific and species-specific analyses, when peak years in article counts were identified, a portion of the articles related to the respective category or species was manually reviewed to examine their context. For the selection of articles to be read, 10 articles were randomly selected from each of the four newspapers for the relevant category or species in a year, resulting in a total of 40 articles. If the number of available articles was fewer than 40, all articles were reviewed. For example, peaks were detected in the 'Resources' category in 2013 and 2014, when there were 180 and 289 articles, respectively (see Results for details). Therefore, 40 articles were randomly selected, and their context was examined. In contrast, peaks were also detected for the 'American eel' in 2012 and 2014, but as there were only 16 articles in each of those years, all articles were read. The reviewed context was categorised and used as supporting information in interpreting the relationship between article surges and the events that occurred in the respective peak years (Table 1).

### Results

### Total number of articles

The annual number of eel-related articles gradually increased, reaching a peak of 977 in 2008, before declining in subsequent years, except for a slight increase observed between 2012 and 2014 (Fig 2). The peak analysis identified 2008

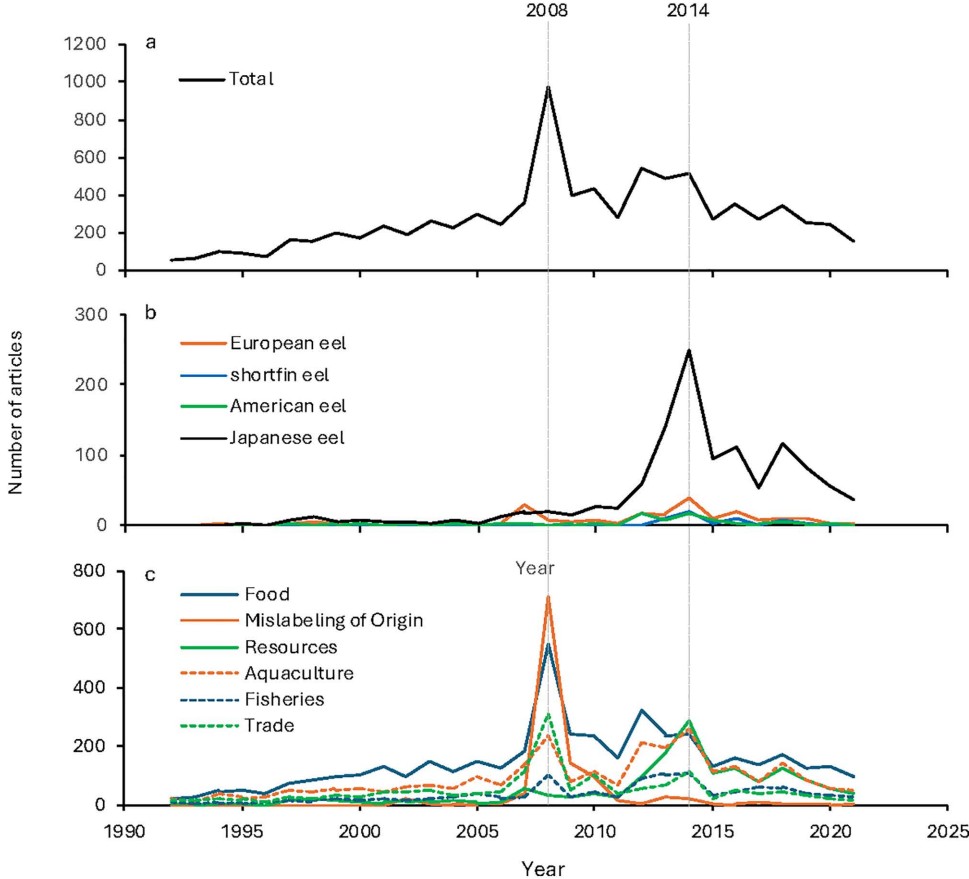

**Fig 2. Number of Japanese newspaper articles related to freshwater eels.** a, total number of articles; b, number of articles on the four eel species consumed in Japan; c, number of articles in the six categories classified based on co-occurrence analysis. Years with prominent peaks are indicated by dashed lines.

as being a year in which the total number of articles relating to eels was significantlly high (Table 4) ($p < 0.05$), while no significant peaks were detected for the other years.

## Category-specific analysis

A co-occurrence network analysis revealed six distinct clusters (Fig 3). Based on the content of the words within each cluster, they were categorised as 'Food', 'Mislabeling of Origin', 'Aquaculture', 'Resources', 'Fisheries', and 'Trade.' Although the 'Food' cluster did not clearly include words related to food (Fig 3), based on it's somewhat close association with *kabayaki* and economic and cultural importance of eels as food in Japan, it appears reasonable to designate a 'Food' category. Using the key words to identify the articles of category (Table 2), the annual number of articles was counted for each category (Fig 2). It is important to clarify that the article categorisation was based on keywords devised from the results of the co-occurrence network analysis, and that the co-occurrence network analysis itself was not directly used for the categorisation.

The peak analysis identified significant deviations in article counts for several categories: 'Food' in 2008; 'Mislabeling of Origin' in 2008, 2009, and 2010; 'Resources' in 2013 and 2014; 'Fisheries' in 2008, 2013, and 2014; and 'Trade' in 2007, 2008, and 2014 ($p < 0.05$) (Table 4). No significant peaks were detected for 'Aquaculture.' In 2008, when the total number

**Table 4. Years in which peaks were detected based on Rosner's ESD test.**

| Analysis type | Species/article category | Years when peaks detected |
|---|---|---|
| Total number of the eel-related articles | | 2008 |
| Species specific | American eel | 2012, 2014 |
| | European eel | 2007, 2014 |
| | Japanese eel | 2014 |
| | Shortfin eel | 2013, 2014, 2016 |
| Category specific | Food | 2008 |
| | Mislabeling of origin | 2008, 2009, 2010 |
| | Resource | 2013, 2014 |
| | Aquaculture | none |
| | Fishery | 2008, 2013, 2014 |
| | Trade | 2007, 2008, 2014 |

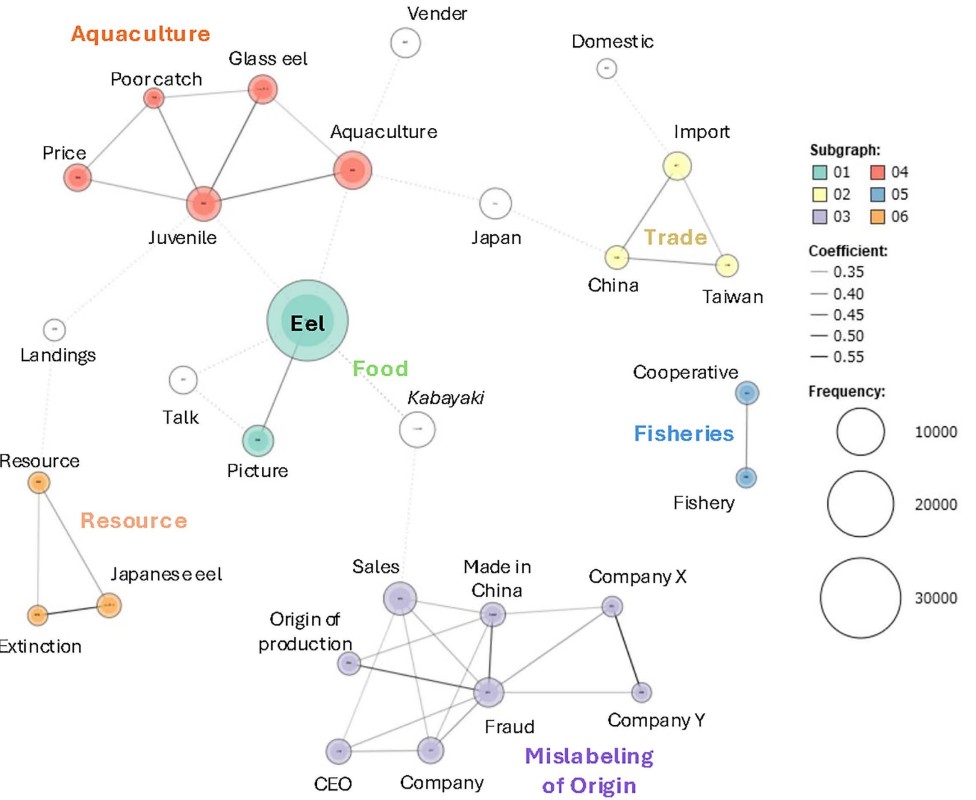

**Fig 3. Results of the co-occurrence analysis of newspaper articles related to freshwater eels.** Different colors indicate distinct clusters representing specific article categories. 'Company X' and 'Company Y' were written in redacted form because they refer to the names of the specific company involved in the mislabeling of origin and its parent company, respectively.

of eel-related articles reached a prominent peak, several businesses were prosecuted for falsely labelling imported eels as domestically produced, making headlines nationwide (Table 1). The number of articles classified under the 'Mislabeling of Origin' category in 2008 was 709, accounting for approximately 70% of the total 977 articles that year (Fig 2). In addition to 'Mislabeling of Origin', the 'Food' and 'Trade' categories also exhibited peak values in 2008 (Table 4). Given that

eels are an important part of Japanese food culture, many articles in the 'Mislabeling of Origin' category likely contained elements related to food. Furthermore, since the mislabeling issue was fundamentally a trade-related problem concerning whether eels were imported or domestically produced, it is reasonable that the 'Trade' category also showed a peak in that year. These results indicate that the primary factor behind the sharp increase in article numbers in 2008 was the issue of mislabeling. The number of articles in the 'Mislabeling of Origin' category also exhibited peak values in 2009 and 2010, with article counts of 140 and 99, respectively, representing a decline compared to 2008 (Fig 2). A review of a part of these articles confirmed that articles from 2009 and 2010 included follow-up reports on the outcomes of court trials, other mislabeling issues, and the impacts of the incidents on the eel market in Japan.

The impact of mislabeling on newspaper coverage in Japan may be related to the significant deviation detected in the 'Trade' category in the previous year, 2007. That year marked the decision to list the European eel in Appendix II of CITES, which likely contributed to the increase in the number of articles classified under the 'Trade' category. In fact, a review of the articles reveals that 32.5% (13/40) of them mentioned CITES. However, a slightly greater proportion—35.0% (14/40)—were reports on the U.S. government's decision in June 2007 to ban the import of eel products from China after detecting the use of malachite green, a prohibited antimicrobial agent. The concern over the safety of Chinese-produced eel may have influenced the widespread media coverage in the following year, 2008, of a major mislabeling scandal in which Chinese-produced eels were falsely sold as domestic produce.

Peak values were detected in 2014 for the 'Resources', 'Fisheries', and 'Trade' categories (Table 4, Fig 2). In that year, the IUCN announced that the Japanese eel had been classified as Endangered on the IUCN Red List, which likely influenced these categories (Table 1). In paticullar, the 'Resources' category includes search keywords such as "extinction" and "conservation" (Table 2), suggesting that a significant number of articles in this category were closely related to the Endangered species designation. A review of the articles categorised under 'Resource', 'Fishery', and 'Trade' in 2014 revealed that the term 'IUCN' appeared in 42.5% of 'Resource' articles (17/40), 40.0% of 'Fishery' articles (16/40), and 57.5% of 'Trade' articles (23/40). These findings suggest that the release of the revised IUCN Red List in 2014 had an influence on newspaper coverage in Japan, particularly within specific thematic categories.

In addition to 2014, peak values were also detected in the 'Resources' and 'Fisheries' categories in 2013 (Table 4). That year, prior to the IUCN's classification of the Japanese eel as Endangered, Japan's Ministry of the Environment designated the species as Endangered on the national red list (Table 1). The significant increase in articles related to 'Resources' and 'Fisheries' in 2013 is highly likely to have been influenced by the Japanese government's designation of the Japanese eel as an Endangered species. These assumptions were also supported by reviewing the content of articles classified into each category at the time. A review of the articles categorised under 'Resource' and 'Fishery' in 2013 revealed that the term 'Ministry of the Environment' appeared in 57.5% of 'Resource' articles (23/40) and 20.0% of 'Fishery' articles (8/40). These findings suggest that the publication of the revised red list by the Ministry of the Environment in 2013 influenced coverage within these categories, particularly in the 'Resource' category.

The GAM analysis examining the impact of category-specific article counts on the total number of articles revealed that 'Food' and 'Trade' had statistically significant effects ($p < 0.001$ and $p < 0.05$, respectively) (Table 5). In particular, 'Food' showed a strong positive association with total article volume ($F = 130.177$). Other categories, including 'Mislabeling of Origin', 'Resources', 'Aquaculture', and 'Fisheries', did not exhibit statistically significant relationships with the total number of articles. The model demonstrated a high explanatory power (adjusted $R^2 = 0.996$, deviance explained = 99.8%), indicating that these six categories account for nearly all the variation in total article count.

### Species-specific analysis

In the species-specific analysis, statistical peak detection identified years in which the number of articles mentioning the American eel, European eel, Japanese eel, and shortfin eel was exceptionally high (Table 4, Fig 2). The results showed that 2014 was a peak year for all four species ($p < 0.05$). Additionally, for the American eel, 2012 was identified as a peak,

**Table 5. Results of Generalized Additive Model (GAM) analysis examining the relationships between the numbers of articles in each category and the total number of articles.**

| Categories | EDF | F value | p value |
|---|---|---|---|
| Food | 1.42 | 130.18 | < 0.0001 |
| Mislabeling of Origin | −0.04 | 0.25 | 0.62 |
| Resource | 0.23 | 1.62 | 0.10 |
| Aquaculture | 0.09 | 0.77 | 0.71 |
| Fishery | 0.21 | 2.10 | 0.53 |
| Trade | 0.52 | 8.08 | < 0.05 |

The response variable was the total number of eel-related articles per year, and the explanatory variables were the numbers of articles in the six categories. The adjusted $R^2$ was 0.996, and the deviance explained was 99.8%.

while for the European eel, 2007 was a peak ($p<0.05$). The Japanese eel had no other peak years apart from 2014 ($p<0.05$), whereas for the shortfin eel, 2013 and 2016 exhibited additional peaks ($p<0.05$). The results of the multiple linear regression analysis indicated that the number of articles for all four species was not significantly correlated with the total number of eel-related articles (Table 6). The adjusted $R^2 = 0.178$ suggests that the model explains only 17.8% of the variance, indicating that species-specific article counts had a limited influence on the overall number of articles.

Possible factors influencing changes in article counts for each species were examined. The year 2014 coincided with the publication of the IUCN Red List assessment results, in which the Japanese eel and American eel were evaluated for the first time and classified as Endangered (Table 1). Additionally, the shortfin eel, previously categorised as Least Concern, was uplisted to Near Threatened. Meanwhile, the European eel remained classified as Critically Endangered, consistent with previous assessments. These findings strongly suggest that the increase in articles related to these four species in 2014 was closely associated with the IUCN Red List assessments. A review of the 2014 articles concerning the American eel, European eel, Japanese eel, and shortfin eel showed that the term 'IUCN' appeared in 53.9% of the articles (41/76). As the first article reporting the IUCN Red List assessment results appeared on 12 June 2014, the subsequent IUCN-related articles were published from mid-June onward. This further suggests that the IUCN Red List assessments of *Anguilla* species influenced the content of Japanese newspaper articles in 2014.

On the other hand, there were cases in which the IUCN Red List assessments did not appear to influence Japanese newspaper coverage. The IUCN Red List assessment of freshwater eel species was subsequently re-evaluated, with

**Table 6. Results of multiple linear regression analysis examining the relationships between the numbers of articles on each species and the total number of articles.**

| Variables | Coefficient | SE | t value | p value |
|---|---|---|---|---|
| Intercept | 211.29 | 38.52 | 5.49 | < 0.0001 |
| American eel | 2.47 | 6.17 | 1.78 | 0.85 |
| European eel | 10.95 | 13.09 | 0.19 | 0.09 |
| Japanese eel | 0.09 | 1.11 | 0.08 | 0.94 |
| shortfin eel | −39.70 | 62.13 | −0.64 | 0.53 |

The response variable was the total number of eel-related articles per year, and the explanatory variables were the annual numbers of articles mentioning each of the four eel species consumed in Japan. No significant relationship was found between the response variable and any explanatory variable. The adjusted $R^2$ was 0.178.

the results for the European eel, Japanese eel, and shortfin eel—whose threat categories remained unchanged—being published in 2020. However, no peaks were detected for any of these species in that year. The re-evaluation results for the American eel were released in 2023, which falls outside the period covered in this study. For 2008, when the European eel was classified as Critically Endangered for the first time (Table 1), despite this year recording an overwhelming peak of 977 for the total number of eel-related articles—1.8 times higher than the second-highest year, 2012 ($n = 540$)—no species exhibited a peak in that year. Furthermore, no significant relationships were observed between the numbers of species-specific articles and total number of eel-related articles (Table 6). These findings suggest that the sudden surge in eel-related articles in 2008 (Fig 2) was not driven by changes in the status of specific species, even though the European eel was designated as Critically Endangered in that year. As shown in the former section (see Category-specific analysis), the factors underlying the sharp increase in total article counts in 2008 are related to mislabeling of eel-origin. Although not identified as a peak, a review of all articles from 2008 in which the term 'European eel' appeared ($n = 19$) found no references to the IUCN Red List, while 78.9% (15/19) included the term 'Mislabeling of Origin', presumably because most of the eels imported from China during that period were thought to be European eels.

CITES-related events may also have influenced newspaper coverage. For the European eel, a peak year was also identified in 2007 (Table 3). This year coincided with the 14th Conference of the Parties (CoP14) of CITES, during which the European eel was listed in Appendix II (Table 1). Based on the article review, we found that 85.7% of the articles (24/28) on the European eel in 2007 focused on this CITES listing, suggesting that this event was a key factor contributing to the increase in media coverage of this species. Another example is the American eel. In addition to 2014, a peak was detected in 2012 for this species (Table 3). In that year, historically poor catches of glass eels used for Japanese eel aquaculture were reported (Table 1), however, a closer examination of the articles on American eels in 2012 revealed that 87.5% of them (14/16) concerned the U.S. Fish & Wildlife Service's consideration of a proposal to list all species of the genus *Anguilla*, including the American eel, in Appendix II of CITES, as well as the eventual withdrawal of that proposal [34]. Based on these findings, CITES is presumed to have influenced the content of Japanese newspaper articles.

Regarding the shortfin eel, 2013, 2014 and 2016 were found to be peak years (Table 4). All articles from 2013 (100%) reported that demand for this species was increasing as a substitute for the Japanese eel and the European eel. In contrast, as noted above, 80.0% of the articles in 2014 mentioned the IUCN, and although this proportion declined to 44.4% (4/9) in 2016, a considerable number of articles still referred to the IUCN Red List context.

## Discussion

### IUCN Red List's influence on media coverage

The findings of this study indicate that, in Japan, the IUCN Red List played a considerable role in shaping newspaper coverage of freshwater eels. In 2014, a year when the IUCN Red List assessments for *Anguilla* species were published [42], the number of articles mentioning the four eel species increased significantly. In the IUCN Red List published in 2014, the Japanese eel was assessed by the IUCN for the first time and classified as Endangered [23]. Although four other species, including the American eel, were also classified as threatened for the first time [7], the number of articles concerning the Japanese eel was substantially higher than that for the other species. One important point to note is that the name '*Nihon Unagi*' (Japanese eel) was formally proposed as the standard Japanese name for *A. japonica* only in 2010 [43]. Although the term '*Nihon Unagi*' had already been in use prior to 2010—for example, a search on Google Scholar using '*Nihon Unagi*' yields over 100 publications between 2000 and 2009. Even considering the circumstances surrounding the name '*Nihon Unagi*,' it is clear that the publication of the IUCN Red List influenced the Japanese newspaper coverage of eels in 2014.

Regarding the context of the articles, the influence of the IUCN Red List was also reflected in a contextual shift in media coverage. Category-specific analysis showed that articles related to 'Resources' peaked in 2014, suggesting that the IUCN Red List assessment contributed to heightened media attention toward eel conservation. Moreover, in 2013,

when Japanese eel was classified as Endangered in the national red list [36], the number of articles in the 'Resource' category also showed the second peak. Prior to 2013 and 2014, articles classified under the 'Resources' category were relatively scarce. However, when the threatened species designation of the Japanese eel occurred in 2013 and 2014, the number of articles in this category increased markedly. This suggests that Red-Listing not only generates more media attention, but also influences the framing of conservation issues, by encouraging a broader discussion of population decline, extinction risk, and resource management. Such a shift in media focus is essential for fostering a deeper public understanding of the ecological challenges facing endangered species.

Taken together, these findings demonstrate that the IUCN Red List influences newspaper coverage of threatened species in Japan, in both quantitative and qualitative ways, indicating that it serves as an important tool for increasing and shaping media reports. By driving quantitative and qualitative changes in media coverage, the IUCN Red List, together with national red lists, may contribute to shaping societal perceptions of conservation issues.

## Limitations of IUCN Red List's media impact

While the IUCN Red List has demonstrated its capacity to influence media coverage, the findings of this study also highlight its limitations. One key issue is that the effect of red-listing on media coverage does not appear to be long-lasting. A previous study found that newspaper coverage of global warming had an immediate, but short-term influence on public concern [39]. Similarly, in this study, the number of articles on the Japanese eel increased significantly following its classification as an Endangered species in 2014. However, when the re-assessment result was published in 2020 [23], no corresponding increase in newspaper coverage was observed. In general, the media tends to report new information as news, making it unlikely for similar content to be repeatedly covered year after year. However, the comparison of peaks in newspaper article counts with events related to eels, such as their designation as threatened species, was possible because the increase in articles triggered by these events did not persist over time. If the designation of a species as threatened had led to a sustained or continuously increasing number of articles, it would not have been possible to detect a distinct peak in a specific year. The usefulness of the method for detecting peaks in article counts is based on the assumption that increases in article numbers do not persist for extended periods. This suggests that while the initial red-listing event attracts attention, its impact diminishes over time, failing to sustain long-term media engagement with conservation issues.

A closer examination of article trends by species and category further illustrates the limited strength of the IUCN Red List's influence on media coverage. The results showed that while the Japanese eel's Endangered classification in 2014 triggered a noticeable response in both species-specific and category-specific reporting, there was no significant peak in the total number of eel-related articles in that year. Moreover, the 'Resources' category, despite its increase in 2014, did not exert a statistically significant effect on the total number of articles. These findings indicate that although the IUCN Red List can influence the framing of conservation discussions in the media, its overall impact on media coverage remains relatively weak. It should be emphasised that, in Japan, eels are regarded as a delicacy and an important part of traditional food culture, consistently attracting media attention. It has been known that large and/or iconic species tend to gain more attention for conservation [44]. Therefore, if even a major species such as the eel can only draw fleeting public attention when classified as Endangered, it is evident that for lesser-known species, which are not widely recognised by the general public, the role that the IUCN Red List can play through media coverage will inevitably be even more limited.

The findings also reveal a lack of media response to Red-Listing events concerning non-native eel species. When the European eel was classified as Critically Endangered in 2008, no significant increase in media coverage was observed in Japanese newspapers. This finding is noteworthy given that one of the major drivers of European eel decline is thought to be consumption in East Asia, especially in Japan [45]. Despite their direct involvement in the consumption of the species, Japanese media appear to remain largely indifferent to conservation concerns related to non-native eels. On the other hand, articles on the European eel exhibited a peak in 2007, the year the species was listed in a CITES appendix [20]. As

more than 80% of these articles referred to the regulation of international trade under CITES, it can be assumed that the media respond when such regulatory developments are expected to affect domestic market dynamics. This assumption is further supported by the increase in articles on the American eel in 2012, when the US government was considering proposing the listing of *Anguilla* species under CITES [34]. Taken together, these findings suggest that the IUCN Red List, which does not directly lead to regulation/restriction and has limited influence on market trends, has relatively little impact on increasing media attention towards non-native species imported and consumed in Japan.

However, in 2014, when the IUCN classified the Japanese eel as a threatened species, there was an increase not only in articles on the Japanese eel, but also on the American eel, European eel, and shortfin eel. This suggests that heightened attention to native species such as the Japanese eel may have contributed to increased interest in non-native species. Further research is needed to explore how media coverage and conservation status of native species may influence public awareness of closely related non-native species.

Overall, these findings suggest that while the IUCN Red List can influence media coverage, its influence is limited in duration, strength, and scope. Based on our research on newspaper coverage in Japan, red-listing events generate immediate media interest, particularly for native species, but do not influence total article number nor sustain long-term engagement. Furthermore, the impact of Red-Listing is largely restricted to species perceived as familiar or locally relevant, with little effect on awareness of non-native species, even when they are directly consumed. This underscores the need for additional mechanisms to maintain public engagement with conservation issues and to foster greater awareness of the global implications of resource use.

## Future directions for conservation and media engagement

While the IUCN has been expanding its conservation frameworks through initiatives such as the Red List of Ecosystems [46] and the Green Status of Species [47], the Red List of Threatened Species remains a valuable tool for raising public awareness. As previous research has revealed that public concern is closely linked to the effectiveness of conservation policies and research efforts [5,48], the role of the IUCN Red List in emphasising public awareness is becoming increasingly important. Although its impact may be limited and short-lived, the findings of this study demonstrate that the IUCN Red List still has the capacity to enhance media coverage of conservation issues.

To improve its effectiveness, experts and conservationists could take a more active role. For example, based on a comparison of threats to deer species in Brazil as portrayed in the media and identified in expert assessments, dos Santos Morini et al. [49] identified gaps that might be addressed through conservation science popularisation and academic-media collaboration. Experts and conservationists could serve as interpreters, ensuring that the significance of red-listing events is effectively communicated to the public. By providing clear and accessible explanations about the ecological implications of species declines, conservation professionals can help sustain public interest beyond the initial announcement of red-listing. Educational initiatives, media engagement, and collaboration with policymakers could further enhance the long-term impact of the IUCN Red List on public awareness.

The limited influence of the IUCN Red List on unfamiliar, non-native species highlights the need for stronger cooperation between countries of origin and importing nations. In cases where species are harvested in one country and consumed in another, conservation efforts should not be the sole responsibility of the exporting nation [50–51]. However, managing natural resources in consumer countries presents challenges, as the ability to regulate imported resources is often constrained. Strengthening management and conservation measures in the countries of origin is therefore critical. The implementation of international agreements, such as the Convention on the Conservation of Migratory Species of Wild Animals (CMS) and the other relevant conventions, could provide additional mechanisms to support sustainable trade and conservation efforts. Enhancing transparency in international supply chains and increasing consumer awareness of the origins and sustainability of imported species could also contribute to more responsible consumption patterns [52–53].

This study revealed that eels primarily appear in Japanese newspaper articles in the context of food. This indicates that eels attract strong public interest as a food item in Japan, suggesting that the desire to "continue eating eels in the future" may serve as a motivation for their conservation. In fact, a recently published study analysing Japanese newspaper articles in 2020 and 2021 suggested that Japanese people can be motivated to conserve eels for consumption [54]. Therefore, improving food-related systems may offer a potential pathway to promote eel conservation. For example, fostering a greater sense of responsibility among consumers in importing nations is crucial. As Japan's Food Labeling Act does not require the specific species of eel to be labelled (https://www.japaneselawtranslation.go.jp/en/laws/view/3649/en), Japanese people are thought to be unaware of the species they consume. Therefore, initiatives that promote consumer education and transparency in food labelling could improve awareness of resource sustainability [55]. Collaborative efforts involving governments, conservation organisations, and the private sector could play a key role in establishing internationally recognised certification systems that ensure sustainable sourcing and responsible trade practices, such as the systems used by the Marine Steward Council (MSC) and Aquaculture Stewardship Councile (ASC) [56–57].

By addressing these challenges through international cooperation, improved consumer awareness, and expert-led communication, the conservation impact of the IUCN Red List can be strengthened. While red-listing alone may not be sufficient to drive long-term media coverage, integrating it into broader conservation strategies could enhance its effectiveness in protecting threatened species and promoting sustainable resource use.

This study found that, in 2014, some events related to the IUCN Red Listing of *Anguilla* species coincided with statistically significant peaks in the number of eel-related articles in Japanese newspapers. However, regression analyses detected no significant correlations between the total number of articles and either resource-related or species-specific articles, indicating that the influence of the IUCN Red List on newspaper coverage in Japan was relatively weak. The influence of the IUCN Red List does not appear to be long-lasting, and this makes statistical peak detection possible. No statistically significant peaks were observed in Japanese newspaper coverage during years when non-native eel species were classified as threatened or near threatened. Overall, these findings suggest that while the IUCN Red List can influence media coverage, its impact is limited in duration, magnitude, and scope. Addressing these limitations could help strengthen the conservation impact of the IUCN Red List, particularly in terms of raising public awareness. Finally, it should be noted that these conclusions should be interpreted with appropriate caution, as certain methodological limitations remain—such as potential biases in keyword selection and the limited consideration of external influencing factors.

## Author contributions

**Conceptualization:** Kenzo Kaifu.

**Data curation:** Hiromi Shiraishi.

**Formal analysis:** Kenzo Kaifu.

**Funding acquisition:** Kenzo Kaifu.

**Investigation:** Kenzo Kaifu.

**Methodology:** Kenzo Kaifu, Hiromi Shiraishi.

**Project administration:** Kenzo Kaifu.

**Resources:** Kenzo Kaifu.

**Software:** Hiromi Shiraishi.

**Visualization:** Kenzo Kaifu.

**Writing – original draft:** Kenzo Kaifu.

**Writing – review & editing:** Kenzo Kaifu, Hiromi Shiraishi, Atsushi Ishii, Aoi Sugimoto.

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
