## [Decision Letter · Decision Letter 0]

15 Jul 2025

PONE-D-25-32104The IUCN Red List and newspaper coverage of threatened freshwater eel species in Japan: a variable but limited influencePLOS ONE

Dear Dr. Kaifu,

Thank you for submitting your manuscript to PLOS ONE. After careful consideration, we feel that it has merit but does not fully meet PLOS ONE’s publication criteria as it currently stands. Therefore, we invite you to submit a revised version of the manuscript that addresses the points raised during the review process.

We look forward to receiving your revised manuscript.

Kind regards,

Mattias Gaglio, PhD

Academic Editor

PLOS ONE

Journal Requirements:

2. Please update your submission to use the PLOS LaTeX template. The template and more information on our requirements for LaTeX submissions can be found at http://journals.plos.org/plosone/s/latex .

“This study was funded by Asahi Glass Foundation, Society for the Promotion of Science (JSPS) KAKENHI Grant Number JP22H00371, and Chuo University.”

“This study was funded by Asahi Glass Foundation, Society for the Promotion of Science (JSPS) KAKENHI Grant Number JP22H00371, and Chuo University.”

“This study was funded by Asahi Glass Foundation, Society for the Promotion of Science (JSPS) KAKENHI Grant Number JP22H00371, and Chuo University.”

“This study analyses the influence of the IUCN Red List on newspaper coverage. At the same time, two of the authors, Kaifu and Shiraishi, are members of the IUCN Species Survival Commission’s Anguillid Eel Specialist Group (AESG) and participated in the Red List assessments of Anguilla species published in 2014 and 2020. While there are no financial conflicts of interest associated with their membership in AESG, their affiliation with the organisation that is the subject of this research is disclosed here for transparency.”

6. We note that you have indicated that there are restrictions to data sharing for this study. PLOS only allows data to be available upon request if there are legal or ethical restrictions on sharing data publicly. For more information on unacceptable data access restrictions, please see http://journals.plos.org/plosone/s/data-availability#loc-unacceptable-data-access-restrictions.

Reviewers' comments:

Reviewer's Responses to Questions

**Comments to the Author**

1. Is the manuscript technically sound, and do the data support the conclusions?

Reviewer #1: Yes

Reviewer #2: Yes

2. Has the statistical analysis been performed appropriately and rigorously? 

Reviewer #1: Yes

Reviewer #2: Yes

3. Have the authors made all data underlying the findings in their manuscript fully available?

Reviewer #1: Yes

Reviewer #2: Yes

4. Is the manuscript presented in an intelligible fashion and written in standard English?

Reviewer #1: Yes

Reviewer #2: Yes

5. Review Comments to the Author

Reviewer #1: This is a good paper and I recommend its publication with some suggestions and minor revision. I appreciate that the authors are looking at eel conservation drivers through popular media (newspapers). They provide a strong narrative through out the paper, explaining or hypothesizing about how article trends are reflected by conservation and fishery phenomenon.

Line 34: You abstract mentions “food and trade dominated media coverage”. You touch on this in the Discussion, but do you think that this points the way to greater public awareness? Should more articles and events be centered around food, as a tool for ultimate conservations since it demonstrates a particular value to eels?

Line 94: What is “production” here? Aquaculture within Japan, or harvested from Japan though raised in foreign aquaculture? All aquaculture eels?

Line 101: Was the topic of eels most frequent among any fish species, of most frequent of any topic, fish-related or otherwise?

Line 121: Was this physical, digital, or both (I assume both)?

Lines 136-165: This section is not bad here, but could be moved to the Introduction adjacent to the material in lines 76-87.

Line 168: Table 1. Just to check, for row 2012, column “Global”, are you talking about US inclusion under CITES, or the US Endangered Species Act?

Line 214: Table 2. Is there any additional explanation as to why many categories have multiple key words, while “Mislabeling of origin” and “Fishery” only have on each?

Line 225: Add “eel” after “shortfin”.

Line 276: Do you mean “each” as in all of the four eel species, or should “each” be replaced with “any”?

Line 289: I keep getting hung up on the term “outlier”. When I see the charts of newspaper articles, these data points strike me as “peaks” or “high points” (as you do in Line 325). The term “outlier” feels like it has a context of being way outside norms, something that has to be “adjusted” for. You are technically correct with your way of using it, but I keep stumbling at it because its used so often.

Line 407: I think this high count of article on “food” and “trade” could be examined a little more in the discussion. Future research or even conservation actions could focus on food and trade, and how demonstrating the value of eels as a food source, and writing about this, could reach many more Japanese people.

Line 413: Does “these categories” refer to the food and trade categories? If so, state that.

Line 457: The main point might be just as strong if you gave one total percentage, not divided into separate species.

Line 532-536: Excellent points.

Line 561-563: Yes, but isn’t that the point, of even the definition, of “news”?

Results & Discussion sections: There is quite a bit of material in the “Results” section could be considered as “Discussion” material because it gets into interpretation and social factors. However, I don’t think you need to re-write this, and in fact I like how you have this blended. Just recognize this is a little different than the “classic” sectioning.

Reviewer #2: This manuscript stands out for its high methodological quality and its original contribution to the study of the societal impact of conservation lists, particularly within the Japanese context and over a span of three decades. The longitudinal approach, combining quantitative and qualitative analyses, allows for an in-depth documentation of the evolution of media coverage and public perception surrounding an emblematic species.

The responses to the previous questions confirm the robustness of this work:

1 : The manuscript is generally technically sound, and the data support the conclusions. The study employs a robust methodology combining quantitative and qualitative analyses over a long period, which contributes to the credibility of the results. However, some methodological limitations—such as potential biases in keyword selection and limited consideration of external influencing factors—should be acknowledged. These do not invalidate the findings but suggest that conclusions should be interpreted with appropriate caution.

2 : The statistical analysis was conducted appropriately and rigorously. The methods used, such as the Rosner test and GAM models, are suitable and carefully applied. The methodological remarks are minor and aim to further enhance robustness without questioning the validity of the results.

3 : In response to PLOS Data policy, the authors indicated that the data used in this study come from archived databases (Asahi Shimbun Cross-Search, Maisaku Database, Nikkei Telecom 21, and Yomidasu Rekishikan) that require a paid subscription. They do not have the authority to make these primary data publicly available due to access restrictions and contractual conditions imposed by the database providers.

Given these legitimate constraints, it is understandable that the underlying data cannot be made fully accessible without restrictions. However, in accordance with PLOS requirements, the authors are expected to provide all derived data, analyses, and results necessary to reproduce the study, as well as clear information on how to access the original databases through the official providers. This approach balances the imperatives of transparency and reproducibility with the limitations imposed by third-party access rights.

4 : As a non-native English speaker, and having already informed you that I use translation tools, assisted writing software, and artificial intelligence to translate the text and draft my review report, I find myself in a position where it is difficult to confidently assess the linguistic quality of the manuscript in English. Although the text appears generally clear and understandable, I am not able to fully evaluate grammatical accuracy, stylistic fluency, or the presence of subtle ambiguities that might require thorough revision by a native speaker or a professional language editor.

Therefore, I recommend that, should the manuscript be accepted, a dedicated language review be considered to ensure that the text fully meets PLOS ONE’s standards for clarity, correctness, and unambiguity. This will help guarantee an optimal presentation of the work to an international readership.

Among the strengths of the work are :

Scientific rigor in data collection and analysis, ensuring the solidity of the conclusions.

Methodological transparency with full data accessibility, facilitating reproducibility and independent verification.

The relevance of the case study choice, which sheds light on ecological, cultural, and economic issues alike.

The originality of the approach, combining advanced text mining tools with fine-grained thematic analysis over the long term.

The concerns raised in the expert report are clearly minor and expressed constructively. They mainly involve:

The opportunity to broaden the thematic lexicon to better reflect conservation issues.

The integration of external variables to refine the analysis of factors influencing media coverage.

Additional verification of certain statistical assumptions.

These suggestions in no way undermine the scientific validity of the work but aim to further enhance the scope and interest of the study. They reflect a commitment to excellence and continuous improvement, and their incorporation can only increase the scientific impact and value of this manuscript.

In summary, this work fully merits publication. The proposed revisions are minor, and their implementation will maximize the clarity, rigor, and scope of the article while highlighting its many intrinsic strengths.

6. PLOS authors have the option to publish the peer review history of their article (what does this mean? ). If published, this will include your full peer review and any attached files.

**Do you want your identity to be public for this peer review?** For information about this choice, including consent withdrawal, please see our Privacy Policy .

Reviewer #1: **Yes: ** Christopher Bowser

Reviewer #2: No

---

## [Author Response · Author response to Decision Letter 1]

23 Jul 2025

Thank you very much for your time and effort in reviewing our manuscript entitled "The IUCN Red List and newspaper coverage of threatened freshwater eel species in Japan: a variable but limited influence." We are sincerely grateful to the editor and reviewers for their thoughtful, constructive, and detailed comments. Your insights have significantly improved the clarity, rigour, and overall quality of our manuscript. We deeply appreciate the careful attention you have given to our work and the opportunity to revise and strengthen the paper in response to your suggestions. Please see the uploaded file for our responses to the individual comments.

---

## [Decision Letter · Decision Letter 1]

18 Aug 2025

The IUCN Red List and newspaper coverage of threatened freshwater eel species in Japan: a variable but limited influence

PONE-D-25-32104R1

Dear Dr. Kaifu,

We’re pleased to inform you that your manuscript has been judged scientifically suitable for publication and will be formally accepted for publication once it meets all outstanding technical requirements.

Kind regards,

Mattias Gaglio, PhD

Academic Editor

PLOS ONE

Additional Editor Comments (optional):

Reviewers' comments:

Reviewer's Responses to Questions

**Comments to the Author**

1. If the authors have adequately addressed your comments raised in a previous round of review and you feel that this manuscript is now acceptable for publication, you may indicate that here to bypass the “Comments to the Author” section, enter your conflict of interest statement in the “Confidential to Editor” section, and submit your "Accept" recommendation.

Reviewer #1: (No Response)

Reviewer #2: All comments have been addressed

2. Is the manuscript technically sound, and do the data support the conclusions?

Reviewer #1: Yes

Reviewer #2: Yes

3. Has the statistical analysis been performed appropriately and rigorously? 

Reviewer #1: Yes

Reviewer #2: Yes

4. Have the authors made all data underlying the findings in their manuscript fully available?

Reviewer #1: Yes

Reviewer #2: Yes

5. Is the manuscript presented in an intelligible fashion and written in standard English?

Reviewer #1: Yes

Reviewer #2: Yes

6. Review Comments to the Author

Reviewer #1: I went through the revised manuscript and I am satisfied that my previous comments and edits were considered and addressed. I appreciate that the authors have both considered the small suggestions as well as added some important points and perspectives to the larger discussions. I have no further suggested edits. I think the paper is well done and an important investigation of science communication for endangered species management.

Reviewer #2: As part of the evaluation of this manuscript submitted to PLOS ONE, I have carefully examined the authors' responses to previous comments as well as the scientific, methodological, and linguistic quality of the article. This report presents my detailed observations and my final recommendation regarding publication.

1.Yes, the authors have satisfactorily addressed all of my comments. The article is scientifically very interesting, and my review focused more on recommendations than on major corrections. Except for a few ambiguities noted in the abstract (see the comments in red in the attached document), which in my opinion do not significantly impact the scientific and methodological quality of the article, the authors have responded to all questions raised and indicated that the recommendations given will be considered in their future studies to improve upcoming publications. Thus, the manuscript now appears suitable for publication.

2.Yes, the manuscript describes a technically rigorous study. The data clearly support the conclusions, and the studies were conducted with rigor, including appropriate and sufficient observations.

3.I had already answered positively to this question in my first report. The statistical analysis was conducted appropriately and rigorously.

4.Regarding data accessibility, the authors took into account my comments from the first report and added the URL of the database in the "Materials and Methods" section (lines 174–179). They noted that although access to this database requires a subscription, readers with access will be able to replicate the analyses presented. This approach appears to comply with PLOS ONE’s data sharing policy.

5.As I have already mentioned, I am not a native English speaker and therefore not qualified to precisely judge the linguistic quality of the manuscript. However, the authors stated that a native English-speaking researcher from the United States proofread the manuscript prior to submission. Additionally, Reviewer 1, who appears to be a native English speaker, pointed out several grammatical issues that were subsequently corrected. Given these points, I consider that the quality of the English in the manuscript has been significantly improved.

Conclusion

At the end of this evaluation, considering the satisfactory responses provided by the authors as well as the scientific, methodological, and statistical soundness of the manuscript, I consider the article to be suitable for publication in PLOS ONE. The final decision on its publication rests with the editor of the journal.

7. PLOS authors have the option to publish the peer review history of their article (what does this mean? ). If published, this will include your full peer review and any attached files.

**Do you want your identity to be public for this peer review?** For information about this choice, including consent withdrawal, please see our Privacy Policy .

Reviewer #1: **Yes: ** Christopher Bowser

Reviewer #2: **Yes: ** DJEZZAR Miliani

---

## [Editor Report · Acceptance letter]

PONE-D-25-32104R1

PLOS ONE

Dear Dr. Kaifu,

I'm pleased to inform you that your manuscript has been deemed suitable for publication in PLOS ONE. Congratulations! Your manuscript is now being handed over to our production team.

Kind regards,

on behalf of

Dr. Mattias Gaglio

Academic Editor

PLOS ONE